# Design and Construction of a Radiochemistry Laboratory and cGMP-Compliant Radiopharmacy Facility

**DOI:** 10.3390/ph17060680

**Published:** 2024-05-25

**Authors:** Angela Asor, Abdullah Metebi, Kylie Smith, Kurt Last, Elaine Strauss, Jinda Fan

**Affiliations:** 1Institute for Quantitative Health Science and Engineering, Michigan State University, East Lansing, MI 48824, USA; asorange@msu.edu (A.A.); metabi@gmail.com (A.M.);; 2Department of Chemistry, Michigan State University, East Lansing, MI 48824, USA; 3Comparative Medicine and Integrative Biology, Michigan State University, East Lansing, MI 48824, USA; 4Radiological Science Department, Taif University, Taif 21944, Saudi Arabia; 5Department of Radiology, Michigan State University, East Lansing, MI 48824, USA; 6WorkingBuildings, Atlanta, GA 30309, USA

**Keywords:** radiopharmacy, radiopharmaceuticals, cleanroom, radiochemistry laboratory, radiation shielding, air quality control, quality control, FDA guidelines, cGMP, translational research, clinical testing

## Abstract

The establishment of a compliant radiopharmacy facility within a university setting is crucial for supporting fundamental and preclinical studies, as well as for the production of high-quality radiopharmaceuticals for clinical testing in human protocols as part of Investigational New Drug (IND) applications that are reviewed and approved by the U.S. Food and Drug Administration (FDA). This manuscript details the design and construction of a 550 ft^2^ facility, which included a radiopharmacy and a radiochemistry laboratory, to support radiopharmaceutical development research and facilitate translational research projects. The facility was designed to meet FDA guidelines for the production of aseptic radiopharmaceuticals in accordance with current good manufacturing practice (cGMP). A modular hard-panel cleanroom was constructed to meet manufacturing classifications set by the International Organization of Standardization (ISO), complete with a gowning room and an anteroom. Two lead-shielded hot cells and two dual-mini hot cells, connected via underground trenches containing shielded conduits, were installed to optimize radioactive material transfer while minimizing personnel radiation exposure. Concrete blocks and lead bricks provided sufficient and cost-effective radiation shielding for the trenches. Air quality was controlled using pre-filters and high-efficiency particulate air (HEPA) filters to meet cleanroom ISO7 (Class 10,000) standards. A laminar-flow biosafety cabinet was installed in the cleanroom for preparation of sterile dose vials. Noteworthy was a laminar-flow insert in the hot cell that provided a shielded laminar-flow sterile environment meeting ISO5 (class 100) standards. The design included the constant control and monitoring of differential air pressures across the cleanroom, anteroom, gowning room, and controlled research space, as well as maintenance of temperature and humidity. The facility was equipped with state-of-the-art equipment for quality control and release testing of radiopharmaceuticals. Administrative controls and standard operating procedures (SOPs) were established to ensure compliance with manufacturing standards and regulatory requirements. Overall, the design and construction of this radiopharmacy facility exemplified a commitment to advancing fundamental, translational, and clinical applications of radiopharmaceutical research within an academic environment.

## 1. Introduction

The role of radiopharmaceuticals in nuclear medicine and molecular imaging has expanded significantly in recent years, with growing applications in both research and clinical settings. Radiopharmaceuticals play a crucial role in diagnostic imaging techniques such as positron emission tomography (PET) and single-photon emission computed tomography (SPECT), as well as in targeted alpha- or beta-particle therapies for various medical conditions, including cancer [1,2,3,4]. As the demand for radiopharmaceuticals continues to rise, the need for specialized facilities capable of producing high-quality radiopharmaceuticals for research and clinical testing becomes increasingly apparent.

In this manuscript, we describe the design and construction of a university radiopharmacy facility designed to support fundamental research, translational research projects, and the manufacturing of radiopharmaceuticals for clinical studies. The facility was developed in accordance with FDA guidelines for the production of aseptic radiopharmaceuticals, adhering to cGMP standards and other United States Pharmacopeia (USP) guidelines [1,3]. The comprehensive infrastructure of the facility, including the radiopharmacy facility and radiochemistry laboratory, is intended to facilitate the preparation, quality control, and release testing of radiopharmaceuticals for PET/SPECT imaging and drugs manufactured for targeted radioligand therapy (RLT) applications.

Section 21, part 211 of the Code of Federal Regulations (21 CFR part 211) describes the minimum standards for the cGMP quality production of drugs at all drug production facilities including not-for-profit organizations and university institutions [5]. Drugs used for PET imaging must follow 21 CFR, part 212, cGMP for Positron Emission Tomography Drugs. The FDA has previously allowed part 212 or USP <823> practices in the manufacturing of radiopharmaceuticals under cGMP that are used in INDs or as part of Radioactive Drug Research Committee (RDRC) approvals [1,6,7].

## 2. Results and Discussion

### 2.1. Facility Design

The design of the radiopharmacy facility was guided by principles of cleanliness, sterility, and radiation safety. An existing room (550 ft^2^) adjacent to a cyclotron vault was allocated for the facility, encompassing both the radiochemistry laboratory and the radiopharmacy (Figure 1). The layout of the facility was carefully planned to optimize workflow efficiency while ensuring compliance with radiation safety and FDA regulatory requirements. The existing room already had an air supply and exhaust that had to be upgraded with an additional chiller, but did not require any new air handling system. 

#### 2.1.1. Basis of Design

The basis of the design was provided by WorkingBuildings, Inc. to meet USP <797>, USP <823>, 21 CFR part 211, and 21 CFR, part 212, standards [8,9]. A modular hard-panel cleanroom system was designed and purchased from CleanAir Solutions, Inc. to house the radiopharmacy operations. The cleanroom system included a gowning room and an anteroom, which served as buffer zones to reduce dust and other contamination from entering the radiopharmacy environment. The gowning room was designed to meet ISO8 (Class 100,000) standards while the anteroom and the cleanroom were designed to meet ISO7 (Class 10,000) standards for particle count parameters, with stringent air quality control measures implemented to maintain cleanliness. Additionally, a laminar-flow biosafety cabinet was installed inside the radiopharmacy to provide an ISO5 (Class 100) sterile working environment and further maintained the clean environment of the radiopharmacy. 

Important facility requirements include the following: temperature < 70 °F (alarm at 67 °F), humidity < 60% (alarm at 57%), maintenance of positive pressure within radiopharmacy (Figure 1), pressure differential alarms, and biannual ISO certification. Additional design considerations included a properly installed coved floor and walls with surfaces that can be cleaned by a variety of aggressive cleaning agents.

##### Compliance 

USP compounding standards are in place to prevent harm to patients that are receiving compounded sterile medications. USP <797> outlines requirements for the sterile compounding facilities and associated standards for operations. USP <823> outlines the standards for the production and compounding of PET drugs for human administration in accordance with state-regulated practice of medicine and pharmacy and approval by Radioactive Drug Research Committee (RDRC) or IND [6,10]. In terms of cGMP, 21 CFR 212 outlines the required production standards and controls for all commercial facilities producing PET drugs irrespective of facility size. Finally, the cGMP principles outline that a PET drug is considered adulterated unless it is produced in accordance with the USP compounding standards and available monographs for PET drugs. All of these standards are in place to ensure that radiopharmaceutical products are prepared aseptically in a cleanroom environment and of suitable quality.

##### A cGMP Radiopharmacy (E20C) vs. a Non-cGMP Radiochemistry Lab (E20) 

The radiopharmacy facility serves as the hub for the preparation and dispensing of radiopharmaceuticals. Preparation of materials intended for administration to human patients is primarily conducted in the cGMP radiopharmacy (E20C), while the non-cGMP radiochemistry laboratory (E20) facilitates radiopharmaceutical quality control processes in addition to fundamental and preclinical research endeavors. Additionally, clinical activities not necessitating a cGMP environment, such as radiosynthesis using synthesis modules in mini hot cells, are conducted in E20, with rigorous environmental monitoring in place. The design and construction of the radiopharmacy required renovation to an existing space, with a modular cleanroom customized to fit the available area and meet ISO standards for certification. Alternatively, the non-cGMP radiochemistry laboratory has one HEPA filter unit installed and is subject to fewer rigid controls. 

#### 2.1.2. Hot Cells and the Radiochemistry Laboratory 

Two lead-shielded hot cells (Von Gahlen), each equipped with a pair of mechanical robot arms, and two dual mini hot cells were installed to facilitate the handling and manipulation of radioactive materials and radiochemistry procedures. The manipulator arms enable manual operation inside the hot cells, as is needed for manual chemistry procedures, splitting F-18 water, and drawing the final dose. The hot cells were strategically located to optimize workflow efficiency, with one hot cell situated in the cleanroom (E20C) and the others in the radiochemistry laboratory (E20). Inside the radiopharmacy hot cell, a laminar-flow insert was installed to provide a sterile laminar-flow environment for radiopharmaceutical handling. The particle counts in the laminar-flow insert met the ISO5 (Class 100) standards. Underground trenches containing shielded conduits were constructed to connect the hot cells and mini hot cells with the cyclotron vaults, enabling the safe and efficient transfer of liquid/gas radioactive materials (RAM). 

#### 2.1.3. Radiation Safety

Radiation safety was a critical consideration in the design of the radiopharmacy facility to minimize undue radiation exposure and protect personnel working within the space [9,11,12]. Radiation safety was provided by physical barriers that included lead-shielded hot cells, dual mini cells for the shielding of radiosynthesis modules, and a shielded trench for transfer of raw radioactive materials. Additionally, radiation monitoring equipment including hand and foot monitors, Geiger counters/survey meters, and three dose calibrators, including one with a sensitive and shielded NaI detector for swipe counting, were strategically placed throughout the facility. Training and standard procedures were put in place as administrative controls to maintain low exposure levels.

##### Hot Cells

The hot cells provide a lead-shielded physical barrier that enables safe handling of RAM within. The three-inch lead walls are sufficient to allow the safe handling of up to 1 Ci of F-18 radioisotopes. The airtight negative pressure configuration of the hot cell retained RAM and prevented volatile RAM elution to the surrounding environment. The hot cells are equipped with manipulator arms for manual operation within, and each hot cell is equipped with a dose calibrator so the amount of RAM can be accurately measured without removal.

##### Trench

The trench, used for transfer of liquid radioactivity between the adjacent hot cells and cyclotron, was shielded with lead bricks and concrete blocks. The concrete blocks (obtained from the hardware store Home Depot) offered a cost-effective and environmentally friendly alternative to lead. The use of concrete blocks also facilitated ease of access to underground trenches while providing adequate radiation protection [13,14].

Various materials were considered for the shielding of gamma-emitting radioisotopes based on the material’s shielding efficiency, a function of the material’s density and mass attenuation coefficient. The mass attenuation coefficient represents the average number of interactions expected to occur between incident photons and a material per unit thickness of said material. Mathematically, this is equivalent to the linear attenuation of the material in question divided by the density of the material. Although the number of expected interactions depends on the material properties and energy of the photons, mass attenuation coefficients for different materials can very similar (Table 1). The shielding material selected for the trench provided sufficient protection from the 511 keV gamma rays expected during F-18 delivery from the cyclotron to the hot cells for radiosynthesis. At the bottom of the trench, 1/2″ diameter PEX tubing served as conduits for the F-18 delivery line of 1/16″ in diameter. A total of 2″ (5.08 mm) of lead bricks provided >10 times of the half-value layer (3.98 mm) for a 511 keV gamma ray. On top of the lead, 14″ (35.56 mm) of concrete blocks added >10 times of the half-value layer (34 mm) for a 511 keV gamma ray (Figure 2). The combination of lead bricks and concrete blocks offered a cost-effective and environmentally friendly way of radiation shielding.

For monoenergetic or monochromatic narrow radiation, gamma-ray absorption is described per the following equation:(1)$$I=I_{0}e^{-(\mu /\rho )(\rho )(x)}\;\mathrm{or}\;I=I_{0}e^{-\mu x}$$

*I*_0_ = original radiation exposure rate (with no shield)

*I* = attenuated radiation exposure rate (with shield)

*μ* = linear attenuation coefficient (cm^−1^) = $\frac{0.693}{x_{1/2}}$

*μ*/*ρ* = mass attenuation coefficient (cm^2^/g)

*ρ* = absorber density (g/cm^3^)

*x*_1/2_ = half-value layer of absorber (cm)

##### Laboratory Personnel Training

Laboratory personnel are required to complete an initial radiation safety training course provided by the institution’s Radiation Safety office, which is an affiliate of Environmental Health and Safety. Successful completion of the course exam with a satisfactory score is necessary to become an approved radiation user. To maintain approval status, annual refresher training is mandatory. Approved users are issued a Landauer collar and ring dosimetry for monitoring radiation exposure to the body and extremities, respectively. Dosimeters are replaced and processed on a monthly basis to monitor and maintain users with exposure levels not exceeding 10% of the occupational dose limits set by the Nuclear Regulatory Commission (NRC). It is critical to adhere to ALARA (As Low as Reasonably Achievable) principles, which involve minimizing exposure time, maximizing distance from the radiation source, and employing appropriate shielding. Tools such as shielded hot cells, automated synthesis modules, manipulator arms, tongs, lead sheets, and lead/concrete blocks support these efforts. Donning appropriate personal protective equipment (PPE) and changing gloves frequently may also help reduce radiation exposure and avoid contamination.

##### Radioactive Material Transfer and Documentation

Comprehensive recordkeeping is employed to document the flow of radioactive materials to and from the radiopharmacy facility. Radioactive materials are received via direct transfer from the neighboring cyclotron, delivery from the adjacent Cardinal Health Radiopharmaceutical worksite, or drop-off from the institutional Radiation Safety office following inspection of vendor-delivered RAM. Following radiochemistry, radiopharmaceuticals may be transferred to another worksite for preclinical or clinical studies. Transport of radioactive materials is facilitated using a two-part containment system (Pinestar Technology, Inc., Jamestown, PA, USA) consisting of a lead pig (1″ shielding) and pig transport case that meets Department of Transportation (DOT) Yellow II Type A packaging requirements (up to 1.5 Ci of F-18) and International Air Transport Association (IATA) Dangerous Goods Regulations. All materials that leave the radiopharmacy are checked for contamination via Geiger counter/survey meter and/or a swipe test of external surfaces prior to departure, including the hands and feet of personnel. A handcart is used to transport the cleared package via designated pathways to minimize the risk of travel accidents leading to exposure or contamination. The radioisotope, total activity level, time of measurement, and container/personnel survey results are documented pre- and post-travel between worksites to ensure the safe flow of materials. For ease of communication, receipt and transport events are recorded in a digital log accessible by radiation users, including the institution’s Radiation Safety Officer (RSO), from any location.

##### Area Surveys

Area surveys are an important aspect of radiation safety and are conducted daily after radiation work is complete. Survey records indicate that the workspace is free of contamination or details the location of remaining (unremovable) contamination, exposure rate, and type of radionuclide on a facility diagram to inform radiation workers of exposure risks. All tools, spaces, and equipment used during radiation work are surveyed using Geiger counter/survey meters or swipe tests to identify activity above background levels once sources have been sequestered. Frequently checking the hands and feet using routinely calibrated survey instruments helps to minimize the transfer of activity to unintended surfaces. Survey records are stored on site and audited monthly by the institution’s radiation safety team to ensure compliance with safety practices. These records serve as a safety tool for communicating with radiopharmacy users, the institution’s RSO, and affiliated stakeholders.

#### 2.1.4. Air Quality Control

Air quality within the facility was maintained through a combination of pre-filters and high-efficiency particulate air (HEPA) filters installed in the ventilation system. The circulated air goes through both a pre-filter and a HEPA filter in one cycle, with pre-filters located at the ground level and HEPA filters installed at the ceiling level of the classified areas. Because the pre-filters trapped larger particles and allowed only fine particles to pass through, they extended the life of HEPA filters. Differential air pressures were monitored and maintained across the cleanroom, anteroom, gowning room, and controlled research space to ensure proper pressure cascades and prevent the dissemination of radiation material as well (Figure 1). The air quality undergoes certification by an independent company every six months. HEPA filters are similarly assessed every six months and replaced if they fail to meet the standards. The laminar-flow insert and biosafety cabinet undergo certification every six months. Additionally, cleanrooms are certified and maintained to ISO8 standards for room E20A, and ISO7 standards for rooms E20B and E20C.

#### 2.1.5. Temperature and Humidity Control

A system for controlling, monitoring, and recording room temperature and humidity was integrated into the facility design to provide optimal conditions for radiopharmaceutical production and quality control testing. Temperature and humidity levels were maintained within specified ranges for compliance and for product stability and integrity. Continuous environmental monitoring was provided, with all data meticulously archived. In the event of any deviation from the defined ranges, notifications were promptly sent to the appropriate facility administrators.

#### 2.1.6. Challenges 

Similar to other complex construction projects, various factors contributed to delays in the completion of the radiopharmacy facility, notably occurring during the COVID-19 pandemic. One significant issue stemmed from disruptions in the supply chain due to COVID-19, exemplified by the delayed delivery of hot cells, which arrived nine months after ordering. Additionally, the construction was confined to a fixed floor plan within the available 550 ft^2^ space, necessitating on-site measurements to ensure the design accurately reflected the actual floor layout. This process revealed discrepancies in the original floor plan, resulting in an unexpected gain of 18 ft^2^ in the cleanroom area, which would have otherwise been overlooked.

Challenges also arose due to structural issues, such as the insufficient thickness of the concrete floor to support the weight of the hot cells, requiring the floor to be cut and replaced with a thicker and reinforced concrete layer. Ineffective communication between construction project stakeholders exacerbated these issues and led to significant delays in updating the construction plans. Further, installation of an incorrect chiller necessitated costly replacements and additional project delays.

The complexity and unique requirements of the project also posed challenges. For instance, while the environmental monitoring system could archive data, it lacked the capability to provide access for retrieving data mandated by FDA regulations. Resolving this issue proved time consuming and contributed to project delays.

The intricacies involved in the construction of a radiopharmacy facility provide ample opportunity for error. Diligent and careful review, coordination, and communication by infrastructure, planning, construction, and management teams are expected to be important for the efficient execution of similar facility construction projects. 

### 2.2. Automated Synthesizer 

The Synthra RN-Plus automated radiosynthesizers, available commercially, fulfill the demand for the flexible automation necessary for the development of radiopharmaceuticals (Figure 3A). These synthesis modules comprise interconnected modular components, including a vial for receiving F-18 in O-18 water from a cyclotron, two reaction vessels equipped with heating/cooling capacity, a vacuum pump for solvent evaporation, a semi-preparative high-performance liquid chromatography (HPLC) purification system with two switchable columns, and a module for reformulating the purified radiopharmaceutical into an injectable solution. Housed within dual mini hot cells, these automated synthesis modules are operated remotely via computer control, executing preprogrammed tasks either manually or being fully automated.

A radiotracer-specific configuration of the synthesis module (Figure 3B) oversees each synthesis step, regulating parameters such as reaction time, temperature, and radioactivity, as well as managing reagent and solvent additions, purification, reformulation, and final product delivery to a sterile vial setup in the laminar-flow insert within the cleanroom hot cell. Throughout the synthesis process, data from various components and detectors are relayed to an interactive computer, allowing radiochemists to monitor real-time information such as temperature, pressure, and radioactivity measurements. This capability enables them to track, trend, and troubleshoot syntheses post-completion.

### 2.3. Administrative Controls, Quality Control Procedures, and Compliance

Administrative controls are in place to uphold facility standards and ensure compliance with regulatory requirements. Regular inspections and audits are carried out to confirm adherence to cGMP standards and FDA guidelines. Training programs have been instituted to equip personnel with the necessary skills and knowledge in facility operations, radiation safety protocols, and radiopharmaceutical production and quality control procedures.

Standard Operating Procedures (SOPs) have been developed and implemented to maintain consistency and reproducibility in manufacturing processes. These procedures encompass preparation, radiolabeling, purification, and formulation of radiopharmaceuticals. Quality control measures, including assessments of appearance, pH, radiochemical purity/yield, chemical purity, radiochemical stability, radionuclide purity, residual solvents, sterility, and bacterial endotoxin levels, have been established in alignment with regulatory requirements and industry standards. A detailed list of SOPs is provided in Table 2. The SOPs are organized into categories such as building and facilities; control components, containers, and closures; equipment; equipment qualification and maintenance; holding and distribution; laboratory controls; packaging and labeling control; personnel qualifications; production and process controls; purchasing and distribution; quality assurance; recall, complaints, and adverse events; and records and reports. Consequently, the SOPs comprehensively address all facets related to the production and quality control of radiopharmaceuticals within the radiopharmacy facility.

### 2.4. Challenges, Limitations, and Future Directions

One of the primary challenges faced during the design and operation of the facility is scalability. As the demand for radiopharmaceuticals grows, the facility must be capable of accommodating increased production volumes without compromising quality or safety. The rapid advancements in radiopharmaceutical technology necessitate regular updates to equipment and procedures. Ensuring compatibility and integration of new technologies with existing systems can pose challenges. Establishing and maintaining a radiopharmacy facility requires substantial financial investment. Ongoing operational costs, equipment maintenance, and regulatory compliance can strain financial resources. Securing funding and optimizing operational efficiency are crucial to ensuring the facility’s long-term sustainability.

Embracing advancements in automation and artificial intelligence can enhance efficiency and accuracy in radiopharmaceutical production and quality control. Integration of robotic systems for sample handling and analysis, and AI-driven predictive maintenance can streamline operations and reduce costs. With evolving regulatory standards, the facility must remain agile and adaptable. Regular training programs and continuous monitoring of regulatory updates are essential to ensure compliance and maintain the facility’s license to operate. Investing in R&D can pave the way for the development of novel radiopharmaceuticals and innovative production techniques. As the facility’s operations grow, there may be a need to expand the infrastructure to accommodate additional equipment and personnel. Strategic planning and phased expansion can help manage growth effectively while minimizing disruptions. Engaging with patients, healthcare providers, regulators, and the community at large can provide valuable insights and foster collaboration. Regular feedback and open communication channels can help align the facility’s objectives with stakeholders’ needs and expectations.

While challenges and limitations are inevitable, proactive planning, continuous improvement, and stakeholder engagement can position the radiopharmacy facility for success in the evolving landscape of nuclear medicine and radiopharmaceuticals. 

## 3. Materials and Methods

### 3.1. Equipment 

The radiopharmacy facility was equipped with state-of-the-art instrumentation and equipment for the production, quality control, and release testing of radiopharmaceuticals.

#### 3.1.1. Radiosynthesizer

The commercially available Synthra RN-Plus automated radiosynthesizers (Synthra, Hamburg, Germany) met the requirement for adaptable automation essential in the creation of radiopharmaceuticals (Figure 3A).

#### 3.1.2. Quality Control (QC)

QC activities vary in complexity from simple tests like thin layer chromatography (TLC) for Tc-99m-labeled kits to comprehensive tests including HPLC, radio-TLC, gas chromatography (GC), and gamma spectrometry for PET radiopharmaceuticals. QC equipment can range from standard analytical instruments like pH meters and HPLC/GC to specialized tools for radioactive sample analysis and endotoxin level monitoring. A complete set of equipment was installed and proper operation was maintained in the radiopharmacy facility to conduct QC and release radiopharmaceuticals for clinical and preclinical use (Table 3). The layout of the equipment in the facility is presented in Figure 4.

Qualification was required for all QC equipment for radiopharmaceutical production, including installation qualification (IQ), operational qualification (OQ), and performance qualification (PQ). 

HPLC/Radio-HPLC: HPLC/radio-HPLC systems consisted of pumps, detectors (including ultraviolet (UV) detectors and radioactivity detectors), and software. OQ tests for detectors included linearity verification and wavelength accuracy. Other tests covered pump accuracy, column oven performance, and autosampler reliability.

Gas Chromatography (GC): GC is often used for residual solvent analysis. IQ protocols were similar to those for radio-HPLC. OQ tests included sensitivity checks for flame ionization detectors and precision and accuracy tests for head space injection systems if equipped.

Radio-TLC: Radio-TLC scanners determine radiochemical purity. IQ follows standard protocols. OQ and PQ tests focus on reproducibility and linearity using the intended radionuclide.

Gamma Spectrometer: Gamma spectrometers are used for identification and radionuclidic purity determination. IQ involved checking documentation and software. OQ tests include energy calibration and efficiency checks. PQ includes reproducibility and linearity tests with intended radionuclides.

Dose Calibrators: IQ for dose calibrators involved verifying documentation and installation conditions. Operational qualification (OQ) tests aim to confirm calibration status through accuracy and reproducibility tests using calibrated radioactivity sources. Performance qualification (PQ) includes accuracy, reproducibility, and linearity tests with intended radionuclides.

#### 3.1.3. Gas Supply

The facility is equipped with three types of gas cylinders: nitrogen, helium, and P-10. Furthermore, compressed air is supplied to the building housing the radiopharmacy facility through a compressor, while hydrogen gas is generated on site by a hydrogen gas generator. Gas pressure is regulated by two-stage regulators and distributed to various equipment via gas manifolds. For instance, helium gas and compressed air are connected to synthesis modules for F-18 drying and pneumatic operation, respectively. Nitrogen gas, hydrogen gas, and compressed air are linked to the GCs, and nitrogen gas and compressed air are available in the hot cells. The P-10 gas is supplied to the gas chamber of the Eckert-Ziegler AR-2000 Radio-TLC scanner. 

#### 3.1.4. Biosafety Cabinet 

The biosafety cabinet (BSC, Class II, Type A2; 1300 Series A2, Thermo Scientific, Waltham, MA, USA) within the cGMP E20C enabled an ISO5 sterile environment for tasks such as media fill and final dose vial setup. Its placement was meticulously selected to accommodate diverse operational requirements within the confined space without compromising particle counts in room E20C. Furthermore, we have clearly delineated the ISO certification status of each area within our facility to offer additional insights into environmental conditions.

#### 3.1.5. Furniture in the Facility

All tables and shelves in the facility were constructed from stainless steel (Metro Stainless Lab Worktable; Metro Super Erecta Solid Shelves, InterMetro Industries Corporation, Wilkes-Barre, PA, USA) and were equipped with special casters (Metro MetroMax iQ Stem Casters, polymer; wheel Tread: polyurethane, antimicrobial and corrosion resistant) that can be locked, offering flexibility for convenient furniture rearrangement during cleaning. Furthermore, these casters are removable and can be autoclaved if necessary, contributing to the maintenance of hygiene standards within the facility. 

#### 3.1.6. Passthrough Hatch

The passthrough hatch (Stainless, 22.75 × 18 inches, Clean Air Products, Minneapolis, MN, USA) linking E20 and E20C offered a convenient means to transfer items between the areas, such as quality control doses in a pig. The air quality in E20C is safeguarded via an interlocking mechanism that prevents simultaneous opening from both sides.

#### 3.1.7. Hot Cells in E20C and E20

The hot cell (Modular Hot Cell with master-slave manipulators, Von Gahlen, Zevenaar, Netherlands) within the cGMP radiopharmacy (E20C) facility served a variety of functions, encompassing cGMP radiosynthesis, RPH formulation, dosing, QC dose preparation, and membrane filter integrity testing. It featured a laminar-flow insert to uphold an ISO5 environment, thereby preserving the integrity of synthesized doses and final dose vials. A second hot cell (Modular Hot Cell with master-slave manipulators, Van Gahlen) was located in E20, with a stacked synthesis hot cell (SB25, Van Gahlen) adjacent and connected to E20C. 

#### 3.1.8. Conduits through Trenches

Conduits were installed to facilitate the transfer of radioactive materials between the E20 dual mini hot cells and both the E20 and E20C hot cells. All hot cells and dual mini hot cells were hermetically sealed and maintained at a negative pressure relative to the surrounding environment. HEPA filters were fitted to the inlet and outlet of all hot cells. The conduits had a diameter of half an inch. A differential pressure of 0.04 inches of water column was maintained between E20C and E20, ensuring a modest driving force. Moreover, not all conduits are operational simultaneously, and any unused conduits were sealed to halt airflow. Additionally, the conduits traversed an underground trench before reaching another hot cell, resulting in a distance longer than 15 feet, further diminishing the likelihood of radioactive material passing through the conduits.

#### 3.1.9. Radiation Monitoring Equipment

Radiation monitoring is critical for the radiopharmacy operation. The radiopharmacy facility is equipped with a survey meter (Model 3 general survey meter with Model 44-9 pancake G-M detector, Ludlum Measurements, Inc., Sweetwater, TX, USA), hand and foot monitor (Model 177 hand and foot monitor, Model 177 Alarm rate meter, with model 44-25 pancake GM hand frisker, and model 44-26 foot frisker, Ludlum Measurements, Inc.), ion chamber (Model 9-3 five range ion chamber, Ludlum Measurements, Inc.) and Na-I well counter as part of the dose calibrator (Capintec, Florham Park, NJ, USA).

## 4. Conclusions

In conclusion, the design and construction of the university-associated radiopharmacy facility represented a significant investment in advancing translational research and clinical applications of radiopharmaceuticals. The associated radiochemistry lab also supports fundamental research. The facility’s comprehensive infrastructure, adherence to regulatory guidelines, and commitment to quality control underscore its pivotal role in supporting innovative research and enhancing patient care in nuclear medicine and molecular imaging. Moving forward, ongoing maintenance and periodic upgrades will be essential to ensure that the facility remains at the forefront of radiopharmaceutical production and quality assurance. The cost for the modular cleanroom, construction, and all equipment described in the manuscript was approximately $2,200,000.

## Data Availability

Data are contained within the article.

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
