# Peer review of "Design and Construction of a Radiochemistry Laboratory and cGMP-Compliant Radiopharmacy Facility"

_pharmaceuticals, 2024, doi:10.3390/ph17060680_

Round 1

Reviewer 1 Report

Comments and Suggestions for Authors

Overall, this manuscript provides a comprehensive overview of the design and construction of a radiopharmacy facility within a university setting. The authors successfully detail the various components and considerations involved in creating a compliant facility to support radiopharmaceutical development research and translational projects.

Suggestions for Improvement:

1. It would be beneficial for the authors to address any challenges faced during the design and construction phases of the facility and offer insights on how these challenges were overcome. Considering the facility's need to handle increased production volumes, it's crucial to consider potential issues such as the interaction between long half-life isotopes and common isotopes within the limited space of the 550 ft2 facility.

2. The layout presented in Figure 1 suggests that only one hot cell is located in the cGMP area, indicating that most production work occurs in the non-cGMP area. Consequently, clarification on the workflow for producing human-use tracers is warranted. Since the facility serves both preclinical and clinical studies within a university context, it would be beneficial to clearly delineate the distinct workflows for preclinical and clinical studies in the non-cGMP area. This clarity will help ensure adherence to regulatory standards and operational efficiency across different study types.

Author Response

  1. It would be beneficial for the authors to address any challenges faced during the design and construction phases of the facility and offer insights on how these challenges were overcome. Considering the facility's need to handle increased production volumes, it's crucial to consider potential issues such as the interaction between long half-life isotopes and common isotopes within the limited space of the 550 ft2facility.

We recognize the importance of addressing the challenges encountered during the design and construction phases of our facility. In our revised manuscript, we have provided a thorough discussion of these challenges and outlined the strategies employed to overcome them. Specifically, we have addressed issues such as space constraints, slab thickness concerns, delays related to COVID-19 supply chain disruptions, and lack of communication caused wrong equipment been installed and removed and replace the proper one. Additionally, we have considered the interaction between long half-life isotopes and common isotopes within our facility's limited space and have taken measures to manage this effectively.

  1. The layout presented in Figure 1 suggests that only one hot cell is located in the cGMP area, indicating that most production work occurs in the non-cGMP area. Consequently, clarification on the workflow for producing human-use tracers is warranted. Since the facility serves both preclinical and clinical studies within a university context, it would be beneficial to clearly delineate the distinct workflows for preclinical and clinical studies in the non-cGMP area. This clarity will help ensure adherence to regulatory standards and operational efficiency across different study types.

Your observation regarding the layout presented in Figure 1 is valid, and we agree that clarification on the workflow for producing human-use tracers is necessary. In our revised manuscript, we have provided a detailed explanation of the workflow for both preclinical and clinical studies, distinguishing between the cGMP and non-cGMP areas. This clarification will ensure adherence to regulatory standards and enhance operational efficiency across different study types.

Reviewer 2 Report

Comments and Suggestions for Authors

This manuscript described the start to finish on setting up a well-designed radiochemistry lab and radiopharmacy in a relatively small space. The authors have thoroughly discussed all aspects of this process, from facility design, regulation compliance, radiation safety, equipment review/layout, process development, etc. It will provide a useful reference source for the nuclear medicine community. However, the authors are recommended to address the following comments/suggestions:

1. Figure 1: BSC is the most essential equipment in cGMP radiopharmacy. However in your layout, BSC is designed close to the entrance (Ante room). This is not for criticizing or "correction-needed" point of view, but it will be great if the authors can discuss whether this design have affected ISO 5 enviroment to some extent, or no differential in partical counts were identified in practice. 

2. What is the purpose of the hot cell in Radiopharmacy, is it to perform cGMP radiosynthesis or only for RPH formulation and dosing?

3. Mark the location of cyclotron on Figure 1.

4. Table 3: Use Molar activity per the nomenclature Guidelines

Author Response

1. Figure 1: BSC is the most essential equipment in cGMP radiopharmacy. However in your layout, BSC is designed close to the entrance (Ante room). This is not for criticizing or "correction-needed" point of view, but it will be great if the authors can discuss whether this design have affected ISO 5 enviroment to some extent, or no differential in partical counts were identified in practice. 

Your observation regarding the placement of the biosafety cabinet (BSC) near the entrance of the cleanroom is valid, and we appreciate your understanding of the practical considerations involved. While this arrangement may seem unconventional, it was carefully chosen to accommodate various operational needs within the limited space available. We have not observed any compromise in particle counts with or without the BSC present in the cleanroom. Additionally, we have clarified the ISO certification status of each area within our facility to provide further context on environmental conditions.

2. What is the purpose of the hot cell in Radiopharmacy, is it to perform cGMP radiosynthesis or only for RPH formulation and dosing?

The hot cell in the cGMP radiopharmacy (E20C) serves multiple functions, including cGMP radiosynthesis, RPH formulation, dosing, QC dose preparation, and membrane filter integrity testing. It is equipped with a laminar flow insert to maintain an ISO5 environment, ensuring the integrity of synthesized doses and the final dose vials.

3. Mark the location of cyclotron on Figure 1.

We acknowledge the importance of clearly marking the cyclotron's location on Figure 1. In the revised version of the manuscript, we have included the cyclotron's position to provide better spatial context.

4. Table 3: Use Molar activity per the nomenclature Guidelines

Thank you for highlighting the need for consistency in nomenclature. We have updated Table 3 to use "molar activity" instead of "specific radioactivity" in accordance with the nomenclature guidelines.

Reviewer 3 Report

Comments and Suggestions for Authors

Review

Design and construction of a radiochemistry laboratory and cGMP-compliant radiopharmacy.

Angela Asor, Abdullah Metebi, Kylie Smith, Kurt Last, Elaine Strauss, Jinda Fan.

The standard and quality of radiopharmaceutical preparations is critical to patient safety and efficacy.  The manuscript describes a non-commercial institution’s development of a radiopharmaceutical laboratory that is compliant with the country’s regulatory requirements. I consider, this work as essential reading to many institutions worldwide who are also considering upgrading or building such facilities.

The manuscript is well written, reads well, covers all of the essential material and is free from grammatical errors.

I have just a few clarifications and comments to ask the authors that could assist readers and enhance to manuscript.

Comments

1.      The authors describe the construction and set-up of a radiopharmacy and a radiopharmaceutical chemistry laboratory, but do not distinguish between the two. From Figure 1, it appears that the radiochemistry laboratory is actually the non-cGMP Radiopharmacy E20 and that the term radiopharmacy refers to  E20C. Could the authors make a brief statement in their introduction where they introduce the facility as to the difference and why the set-up as drawn. It is assumed that only  non-clinical work will be conducted in E20. Is this correct?

2.      Can they make a comment on the airflow at the ‘pass thru’ hatch in meeting air quality standards in E20C.

3.      There seems to be a considerable amount of ‘furniture in E20C- the cGMP Radiopharmacy’, e.g. a shelf and 2 3FT Tables. Have the author’s anticipated the effects these would have during cleaning and maintenance of the room to cGMP standards. Mot clean rooms are constructed with minimum fit out and essential materials are brought into the clean room to minimise clutter, accumulation of particulate matter, enhance cleaning and hence maintain air quality. For example, are the tables mobile to facilitate cleaning? Could these be replaced by mobile stainless-steel trolleys? Does the ‘shelf’ and whatever is stored on them pose issues with particulates, air-quality and cleaning over time?

4.      It is assumed that there is no transfer between the hot-cells in E20 to the one in E20C, is this correct? Could the authors clarify this.

5.      Have the authors considered periodic testing of transfer lines for microbial growth between the cyclotron and the hot-cell in E20C.

6.      The authors described the underfloor trench between the hot-cells. The radiation shielding aspects have been well described but what about risks to air quality and potential contamination?  Although, the E20C lab is under positive pressure- has the potential risks from this been considered?

7.      The maintenance of air-quality was well descried in section 2.4, But can the authors describe how air quality is measured and how frequent? Or is there continuous monitoring? What about air quality monitoring/ measurement and recording in hot-cell or laminar flow?

8.      Could the authors describe the purpose of the Laminar flow cabinet in the cGMP E20C Radiopharmacy laboratory. Is this permitted and appropriate in this laboratory? Has its impact been assessed in terms of air quality, cleaning etc?

Author Response

  1. The authors describe the construction and set-up of a radiopharmacy and a radiopharmaceutical chemistry laboratory, but do not distinguish between the two. From Figure 1, it appears that the radiochemistry laboratory is actually the non-cGMP Radiopharmacy E20 and that the term radiopharmacy refers to  E20C. Could the authors make a brief statement in their introduction where they introduce the facility as to the difference and why the set-up as drawn. It is assumed that only  non-clinical work will be conducted in E20. Is this correct?

We acknowledge the need for clarification regarding the distinction between the radiochemistry laboratory and the radiopharmacy. In this manuscript, radiopharmacy refers to the facility where radiopharmaceuticals are prepared and dispensed, including the cGMP radiopharmacy E20C and part of non-cGMP radiopharmacy E20. Quality control of radiopharmaceuticals is performed in E20. The radiochemistry laboratory is the place where research can be conducted on the development of radiochemistry and radiopharmaceuticals, including E20 and possibly the cleanroom following all regulations when it is available.

The design and construction of the radiopharamcy facility was a renovation of the existing space. A modular cleanroom was customized to fit the available space. E20 supports the quality control of radiopharmaceuticals, fundamental and preclinical research work. The clinical work that does not requires cGMP environment can be performed in E20 as well, such as the radiosynthesis on the synthesis modules in the mini hot-cells, in which the environment was monitored.

  1. Can they make a comment on the airflow at the ‘pass thru’ hatch in meeting air quality standards in E20C.

The pass-through hatch is designed with an interlock to prevent simultaneous opening from both sides, ensuring no compromise in air quality in E20C.

  1. There seems to be a considerable amount of ‘furniture in E20C- the cGMP Radiopharmacy’, e.g. a shelf and 2 3FT Tables. Have the author’s anticipated the effects these would have during cleaning and maintenance of the room to cGMP standards. Mot clean rooms are constructed with minimum fit out and essential materials are brought into the clean room to minimise clutter, accumulation of particulate matter, enhance cleaning and hence maintain air quality. For example, are the tables mobile to facilitate cleaning? Could these be replaced by mobile stainless-steel trolleys? Does the ‘shelf’ and whatever is stored on them pose issues with particulates, air-quality and cleaning over time?

All furniture in E20C is made of stainless steel and equipped with special castors for easy removal and autoclaving to maintain cleanliness and air quality. The tables and shelf provide cleanable surfaces to support the setup and handling of protocols involved in preparing and dispensing radiopharmaceuticals.

  1. It is assumed that there is no transfer between the hot-cells in E20 to the one in E20C, is this correct? Could the authors clarify this.

No conduits were installed between the E20 hot cell and the E20C hot cell. However, conduits were installed between the E20 dual-mini hot cells and both hot cells, allowing transfer of radioactive materials between mini-cells and hot cells.

  1. Have the authors considered periodic testing of transfer lines for microbial growth between the cyclotron and the hot-cell in E20C.

Your suggestion regarding periodic testing of transfer lines for microbial growth is valuable. We considered flushing the lines with 70% isopropanol weekly and replacing them quarterly to minimize microbial growth.

  1. The authors described the underfloor trench between the hot-cells. The radiation shielding aspects have been well described but what about risks to air quality and potential contamination?  Although, the E20C lab is under positive pressure- has the potential risks from this been considered?

We appreciate your concern. The setup minimizes the risk of contamination. All hot cells and dual-mini hot-cells are air tight and negatively pressured to the environment. All hot cells’ inlet and outlet equipped with HEPA filters. The conduits have a diameter of half inches. The differential pressure between E20C and E20 is maintained at 0.04 inches of water column. Thus, the driving force is small with this pressure difference. Additionally, not all conduits are in use at the same time, unused conduits can be sealed to stop air flow through. Furthermore, because the conduits first go down to the underground trench, then go back to another hot cell, the distance is longer than 15 feet, further reduce the possibility of RAM goes through the conduits. In operation, we have not observed any radiation contamination through the conduits yet. We will continue observe and record.     

  1. The maintenance of air-quality was well descried in section 2.4, But can the authors describe how air quality is measured and how frequent? Or is there continuous monitoring? What about air quality monitoring/ measurement and recording in hot-cell or laminar flow?

Environment is monitored in the facility, but not the particle counts. The quality of the air is certified by an independent company every six months. HEPA filters also qualified every six-month, replaced if failed. The laminar flow insert and the biosafety cabinet are certified every year.

  1. Could the authors describe the purpose of the Laminar flow cabinet in the cGMP E20C Radiopharmacy laboratory. Is this permitted and appropriate in this laboratory? Has its impact been assessed in terms of air quality, cleaning etc?

Thank you for raise this concern. The biosafety cabinet in the cGMP E20C provide ISO5 sterile environment for media fill and final dose vial setup. The BSC has its own HEPA filter, it may help drop the particle counts in the cleanroom, we did not observe any compromise of the particle counts with or without the BSC present in the cleanroom. The cleaning of the biosafety cabinet can be accomplished with the help of an extension rod. The impact of BSC on air quality and cleanliness has been assessed and observed to be effective.
